# Selection, Ignorability and Challenges with Causal Fairness

**Jake Fawkes**                                                   JAKE.FAWKES@STATS.OX.AC.UK
*Department of Statistics, University of Oxford*

**Robin J. Evans**                                                    EVANS@STATS.OX.AC.UK
*Department of Statistics, University of Oxford*

**Dino Sejdinovic**                                           DINO.SEJDINOVIC@STATS.OX.AC.UK
*Department of Statistics, University of Oxford*

**Editors:** Bernhard Schölkopf, Caroline Uhler and Kun Zhang

## Abstract

In this paper we look at popular fairness methods that use causal counterfactuals. These methods capture the intuitive notion that a prediction is fair if it coincides with the prediction that would have been made if someone's race, gender or religion were counterfactually different. In order to achieve this, we must have causal models that are able to capture what someone would be like if we were to counterfactually change these traits. However, we argue that any model that can do this must lie outside the particularly well behaved class that is commonly considered in the fairness literature. This is because in fairness settings, models in this class entail a particularly strong causal assumption, normally only seen in a randomised controlled trial. We argue that in general this is unlikely to hold. Furthermore, we show in many cases it can be explicitly rejected due to the fact that samples are selected from a wider population. We show this creates difficulties for counterfactual fairness as well as for the application of more general causal fairness methods.

## 1. Introduction

Recently there has been a large body of work on the problem of fair machine learning. This has stemmed from concerns that training data often contains human and societal biases that can be replicated by machine learning models, causing unfair treatment to certain groups on the basis of protected attributes such as race, gender and disabilities. This has given rise to a large variety of statistical fairness definitions such as demographic parity (Feldman et al., 2015), equality of opportunity (Hardt et al., 2016), fairness through awareness (Dwork et al., 2012) and many more (Verma and Rubin, 2018). Following this there have been many different approaches to achieve these definitions, such as variational inference (Louizos et al., 2015), adversarial learning (Zhang et al., 2018) and optimal transport (Chiappa et al., 2020).

Following results showing many statistical fairness definitions are mutually incompatible (Kleinberg et al., 2016; Pleiss et al., 2017), and so can not be simultaneously satisfied apart from in trivial scenarios, new definitions were proposed based on causality (Kilbertus et al., 2017; Zhang and Bareinboim, 2018; Kusner et al., 2017; Nabi and Shpitser, 2018; Chiappa, 2019). This work argues that causal definitions using interventions and counterfactuals capture a more intuitive and correct understanding of what it means for an algorithm to

be fair, and that only by understanding the causal relationships in our data can we hope to satisfy fairness (Chiappa and Isaac, 2018; Loftus et al., 2018).

In this paper we focus on the most popular causal fairness definitions, which use causal counterfactuals (Kusner et al., 2017; Nabi and Shpitser, 2018; Chiappa, 2019). Causal counterfactuals aim to answer questions of the form "what would have happened to $Y$ had $X$ been different, given we hold anything that doesn't depend on $X$ constant?". In fairness settings the counterfactuals are based on what would have happened had the value of a sensitive attribute been different, given we hold all other background conditions constant. Counterfactual Fairness (Kusner et al., 2017) says our predictions are fair for an individual if they align with those in a counterfactual world in which their sensitive attribute had been different. For example, a prediction of the probability that a woman defaults on her loan is fair if it coincides with the prediction they would receive if they had they been counterfactually born a man, given everything else is held constant.

To achieve this requires a causal model to be fitted to data. This model allows us to compute approximate counterfactuals which our model is fair in relation to. Therefore, it is critically important that the class of causal models we search over contains at least one model with the correct counterfactuals. The most common class in causal literature is the class of models with independent noise. Informally, this assumes that our factual data and the counterfactuals can be described by a set of deterministic equations with the addition of random noise that is not correlated with anything else.

In this paper we challenge this in a fairness context. Our argument rests on the fact that if you assume this, the approximate counterfactuals generated by these models have properties you would only expect to see in a randomised controlled trial, and so seem implausible. Moreover, data in fairness problems is usually selected in some way. So, we show that if an independent model could fit the general population the faithfulness assumption inherent to graphical models suggests that none could fit the selected population. Hence, the noise variables effectively must be dependent.

We argue this creates problems for achieving counterfactual fairness and for causal fairness more generally. This is because correctly fitting models with dependent noise is considerably more challenging as we do not know the correct nature of the dependency and cannot tell without more data. It also means that the modelling assumptions required for many methods from the field of causality do not hold. We give an explicit example of this in the case of path-specifc fairness.

## 1.1. Paper Outline

In Section 3 we introduce the ignorabillity assumption which is key to our argument. We discuss it's relevance to a randomised controlled trial, how it arises from common modelling assumptions in the fairness literature and why it is unlikely to hold in practice. Following this in Section 4 we lay out an explicit causal argument against ignorability. This leads to conditions for when we can assume an independent noise model and a constraint which can show no independent noise model fits. Finally in Section 5 we discuss the difficulties this raises for causal fairness.

## 2. Preliminaries

### 2.1. Notation and Definitions

#### 2.1.1. CAUSAL DEFINITIONS

Following Pearl (2009) and Peters et al. (2017) a *Structural Causal Model* (SCM) $\mathcal{M} = \langle U, V, F, P(U) \rangle$ consists of:

- $U$, a set of *noise variables* or latent background variables; these are factors not caused by any variable in the set $V$ of *observable variables*.

- $F$, a set of *structural equations* $\{f_1, \ldots, f_n\}$, one for each $V_i \in V$, such that $V_j = f_j(pa_j, U_j)$, $pa_j \subseteq V \setminus \{V_j\}$ and $U_j \subseteq U$ where $pa_j$ is notation for the parents of $V_i$. This notation comes from the fact that the model gives rise to a causal graph which we assume to be a DAG.

- A *probability distribution* $P(U)$ over the latent variables $U$.

We may model the distribution of a set $Z$ following an intervention on a subset of the other variables $W \subseteq V \setminus Z$, by replacing the structural equation for each $W_i \in W$ by the fixed value $W_i = w_i$. We use the potential outcome notation, so $Z(w)$ is a random variable that has distribution of $Z$ after we have intervened to set $W = w$. Further the SCM allows for the computation of *structural counterfactuals*. That is, for an individual with background variables $U = u$, the structural counterfactual for $Z$ given $W = w$ is denoted by $Z(w, u)$ and is the unique solution for $Z$ given $U = u$ and by replacing the equations for $W$ with the fixed value $W = w$. We often omit the $u$ when it is clear from context and just write $Z(w)$.

Given our probability distribution, $P(u)$, we can infer the distributions over our structural counterfactuals given evidence. That is, we can compute $P(Z(w) = z \mid E = e)$ by finding the posterior distribution for $U$ given $E = e$, substituting $W = w$ in our equations, and then using our posterior distribution $P(U \mid E = e)$ to give the probability in question. The evidence, $E$, could be something counterfactual; for example, we might have observed $Z = z', W = w'$ and then want to compute the probability that $Z = z$ in a counterfactual world in which $W$ is fixed to the value $w$.

We refer to these as structural counterfactuals in order to emphasise that they are counterfactuals from a structural causal model. This does not mean they are truly counterfactuals and generally we can only give them this interpretation when the causal model is suitably motivated by our beliefs about the true state of the world. Whenever we make the assumption that there is a true causal model that generates our data we will denote it by $\mathcal{M}^*$. Whenever a true causal model $\mathcal{M}^*$ is assumed, we will use $V^*(a)$ to denote the counterfactual of $V$ according to this causal model.

The key focus of this paper is whether or not it is valid to assume that the noise variables, $U$, are jointly independent in fairness settings. This assumption is commonplace in the wider causal literature (Peters et al., 2017) and many key results rely on it, the most obvious being that d-separation implies conditional independence.

### 2.1.2. COUNTERFACTUAL FAIRNESS

Now we introduce counterfactual fairness (Kusner et al., 2017). First in the general fairness setup we suppose we have access to a dataset $\Delta = \{a^n, x^n, y^n\}_{n=1}^{N}$ of individuals where $a^n$ indicates the *sensitive attributes*, $x^n$ is a list of covariates and $y^n$ is some outcome of interest we wish to predict. We want to form a predictor $\widehat{Y}$ which is not discriminatory on the basis of our sensitive attributes, this is known as being fair'. In order to do this we need some definition of fairness. For counterfactual fairness we assume that there is some true causal model of the world $\mathcal{M}^*$ and that relative to this model a predictor $\widehat{Y}$ is *counterfactually fair* if for all contexts $X = x, A = a$ we have

$$P\left(\widehat{Y}(a) = y \mid X = x, A = a\right) = P\left(\widehat{Y}(a') = y \mid X = x, A = a\right) \tag{1}$$

for all values $y$ and $a'$.

In order to achieve counterfactual fairness, we fit some causal model $\mathcal{M}$ aiming to approximate $\mathcal{M}^*$. We then either use noise variables arising from $\mathcal{M}$ and covariates that are not causally dependent on $A$ as inputs to $\widehat{Y}$, or use $\mathcal{M}$ to generate structural counterfactuals and use regularisation to enforce that the predictor outputs the same value on the potential outcomes as in their observed values (Russell et al., 2017). Kusner et al. (2017) emphasise that the causal model $\mathcal{M}$ we fit must be suitably causally motivated for us to expect any predictor formed in this way to be counterfactually fair, relative to the true causal model $\mathcal{M}^*$. Therefore we would hope there exists some causal model in the space we search over that has the correct counterfactuals. That is, the structural counterfactuals align with these 'true counterfactuals' from $\mathcal{M}^*$.

## 2.2. Introducing the Law School Example

We use one of the main examples from Kusner et al. (2017) throughout to explain our ideas. They aim to form a predictor for US law school admissions, which should be fair with respect to the sensitive attributes race and sex. To train the predictor we have data from people who previously attended law school. We have their college GPA and LSAT scores, as well as their sensitive attributes race and sex. From here we aim to impute their first year average grade (FYA). We would then use this to predict if a new applicant would succeed at law school and so if they should be admitted or not. Kusner et al. (2017) assume the observed variables follow the causal DAG in Figure 1; whilst they use different causal models to estimate the noise variables, this structure over the observed variables remains constant.

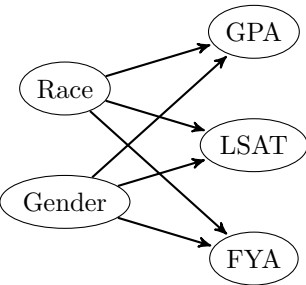

Figure 1: A causal DAG for the Law School example

## 2.3. Ancestral Closure of the Sensitive Attributes

When drawing these causal graphs to describe our data, a common assumption is that the set of sensitive attributes is *ancestrally closed*. By this we mean it has no observable cause or unobserved cause that is shared with another variable. This makes sense in many scenarios;

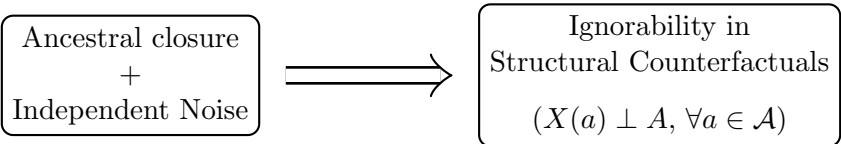

for example, nothing could be said to cause someone's gender or many disabilities. In the case of counterfactual fairness ancestral closure is mentioned as an explicit requirement on the set of sensitive attributes (Kusner et al., 2017). This is because otherwise we could discriminate on the basis of things that cause our sensitive attribute. Kusner et al. (2017) give the example that we could discriminate on the basis of mother's race if only race was sensitive and we did not enforce ancestral closure. Therefore given the fact that ancestral closure can be seen as a requirement and is also assumed in most DAGs we can find in the literature (Kusner et al., 2017; Russell et al., 2017; Kilbertus et al., 2020; Nabi and Shpitser, 2018; Chiappa, 2019), we make this assumption throughout.

## 3. Ignorability

As stated, our main focus will be on whether it is reasonable to assume that the noise variables, $U$, are mutually independent in fairness settings. The difficulty comes from the fact that independent noise and ancestral closure together imply that our structural counterfactuals satisfy *ignorability*[1]. This assumption is more commonly seen in the context of randomised controlled trials and it is as follows:

$$X(a) \perp A, \qquad \forall a \in \mathcal{A}. \tag{2}$$

This implies that for all $a, a'$:

$$\left(X \mid \{A = a\}\right) \stackrel{d}{=} \left(X(a) \mid \{A = a\}\right) \stackrel{d}{=} \left(X(a) \mid \{A = a'\}\right). \tag{3}$$

Imagined in a randomised control trial where $A$ is now our treatment and $X$ is the measured covariates, this says if we want to know how the untreated group would look had we counterfactually given them the treatment then we only need to look at what happened in the treated group. That is, due to the randomisation of the treatment, those individuals in the untreated group would (on average) look like individuals in treated group, had we counterfactually chosen to treat them instead. This is what allows us to estimate the effect of a treatment in a randomised controlled trial by the difference in means between the treated and untreated groups. Ignorability is therefore a very strong assumption in general, the fact that know we it is satisfied in randomised controlled trials is what makes them the 'gold standard' of causal inference.

In fairness settings where there is no such motivation from randomisation of 'treatment', ignorability seems like a much stronger assumption. In the context of the law school example with counterfactuals relating to sex, this would mean if the group of males that applied were counterfactually born female, they would look like the group of applying females.

---

1. Also known as exchangeability.

This implies that for every male that applied, if they had counterfactually been born female then they would have attended college to get a GPA, taken the LSAT, and that their GPA and LSAT grades would be indistinguishable from the grades attained in the real world by the women who applied to law school.

It seems natural to be concerned that this will not hold. We might feel our society unfairly pushes women away from considering a career in law, and so believe if some men who applied for law school had been born female instead they would have been deterred from taking the LSAT, for example. If we imagined the same scenario in the 1950s, when there was poor access to higher education for many women, we would almost certainly expect that most men who attend college would not have attended college if they had been born female and therefore would not have a GPA. If our structural counterfactuals are not at all similar to what we would expect a true counterfactual to be like in this extreme case, it is hard to see how can we have confidence that they resemble true counterfactuals when applied to other fairness scenarios.

Furthermore, we can imagine scenarios when treating counterfactuals like this could be intuitively very unfair. As an example, again we look at law school admissions, but now our sensitive attribute $A$ is the presence of a particular disability with severe adverse effects; for instance, it might mean that, on average, sufferers can only work or study for half the amount of time per day than someone who does not have this disability. If an individual were able to attend college to get a GPA, take the LSAT, and perform well enough in both of these to apply to law school *despite* having this disability, it is reasonable to assume that they are an exceptional candidate and would have performed exceptionally well relative to all candidates had they been born without any disability. However, their structural counterfactuals formed as above would look like an average applicant born without the disability. Therefore a predictor which is counterfactually fair relative to these structural counterfactuals would simply treat this candidate as an average applicant without the disability. This does not capture an intuitive notion of fairness in this scenario. Further it does not align with what we imagine counterfactual fairness as doing. The structural counterfactuals are failing to correcting for the difficulties of having this sensitive attribute, which is one of the main appeals and claims of counterfactual fairness.

We note that almost all models we found in the literature on fairness using causal counterfactuals assumes a causal model that is both ancestrally closed and has independent noise variables, either explicitly (Kusner et al., 2017; Russell et al., 2017; Kilbertus et al., 2020) or implicitly for identification results (Nabi and Shpitser, 2018; Chiappa, 2019). We now give more detailed analysis of when this may seem to be a reasonable or unreasonable assumption.

## 4. Selection

### 4.1. Theoretical Results

In order to formalise the issues raised in the previous section we cast it as a problem due to selection from a wider population. Key to our analysis is that in this population we are comfortable with the assumption that the 'true' counterfactuals satisfy ignorability.

Therefore we focus our analysis on birth sex and take the wider population to be the general population of, for example, a country. Now ignorability does not seem such a strong assumption as birth sex is random and there is no selection whatsoever, so we can loosely imagine this as a large randomised trial[2]. This allows us to make the following assumption about the nature of the data generating process:

**Assumption 1** *There is some true causal model $\mathcal{M}^*$ that generates our covariates, $X$, for the entire population. In $\mathcal{M}^*$ the noise variables are independent and in the causal DAG following from $\mathcal{M}^*$ the sensitive attribute set is ancestrally closed. Further, we allow the domain of $X$ to be expanded so that we write $X_j = \emptyset$ if an individual does not possess the jth covariate.*

We denote the structural counterfactuals arising from $\mathcal{M}^*$ by $X^*(a)$ and call these the *true counterfactuals*. Due to the form of $\mathcal{M}^*$ the counterfactuals will satisfy $X^*(a) \perp A$; however as we stated, this is a more comfortable assumption in the general population. It is important to note we do not consider race here, as the assumption of ignorability or behaving like a randomised control trial seems much more unreasonable. We discuss this further in Appendix **??**. However it should be said that if whenever it is assumed that an SCM with independent noise in which race is ancestrally closed can fit the counterfactuals we make the same assumption of ignorability and this is still unlikely to hold.

Continuing with the example of the law school predictor our covariates $X$ are GPA and LSAT and we use GPA $= \emptyset$ to indicate when an individual in the general population has not completed college and so lacks a GPA. As above we focus our analysis on birth sex, and assume $A$ can only take two values $a$, $a'$; we do this because it is consistent with the measurements used in many of the datasets we will look at.

We use a binary variable $S$ to indicate if an individual lies in the dataset we have access to. For example, in the law school example $S = 1$ if an individual applied to law school. Now to try to achieve counterfactual fairness with our dataset we would be fitting a causal model $\mathcal{M}$ on those with $S = 1$. As discussed in Section 3 many models in the causal fairness literature fall into the following class:

**Definition 1** *Let $\mathbb{M}_{S=1}$ be the set of causal models that fit the data, in which all noise variables are jointly independent and further, give rise to a DAG in which the set of sensitive attributes is ancestrally closed.*

The question is: when does there exist a model $\mathcal{M} \in \mathbb{M}_{S=1}$ such that the structural counterfactuals of $\mathcal{M}$ align with the true counterfactuals from $\mathcal{M}^*$? The following gives a characterisation of this:

**Proposition 2** *There exists an $\mathcal{M} \in \mathbb{M}_{S=1}$ such that the structural counterfactuals from $\mathcal{M}$ align with the true counterfactuals from $\mathcal{M}^*$ if and only if we have:*

$$X^*(a) \perp A \mid S = 1 \ , \ \forall a \in A. \tag{4}$$

---

2. We note that even still this is an approximation and at most there would be near ignorability. We take this assumption exactly simply to allow us to make some formal analysis. However rejecting this assumption in general supports our argument as it shows in no population should a practitioner be happy with ignorability.

That is, if we satisfy *ignorability under selection*. Furthermore, we can state this in terms of a constraint on selection which resembles a scaled version of counterfactual fairness:

**Proposition 3** *We have ignorability under selection if and only if selection satisfies the following:*

$$\frac{P(S(a) = 1 \mid X = x, A = a)}{P(S = 1 \mid A = a)} = \frac{P(S(a') = 1 \mid X = x, A = a)}{P(S = 1 \mid A = a')}$$

*for all $a, a'$ and $x$ such that $P(X = x \mid A = a) > 0$.*

Therefore if either of these conditions are violated we should not expect any model in this class to capture the correct counterfactuals. This is clearly a problem if the justification of our causal fairness method relies on it being able to—in principal—capture the correct counterfactuals.

## 4.2. When will Ignorability Under Selection Hold?

We now look further at when we can expect ignorability under selection to hold and therefore when we can fit or assume a model in $\mathbb{M}_{S=1}$. In order to do so we first introduce the definition of faithfulness:

**Definition 4** *A distribution $P(V)$ is* faithful *with respect to some graph $\mathcal{G}$ if whenever we have $A \perp B \mid C$ for sets of variables $A, B, C$ then $A$ is d-separated from $B$ by $C$ in $\mathcal{G}$. That is any conditional independences are implied by d-separation.*

Faithfulness is the converse to the statement that d-separation implies conditional independence and is commonplace in the causal inference literature. A violation of faithfulness entails an exact balancing out of causal effects so that they leave no probabilistic trace and the assumption is often justified by arguing that this is unlikely to occur in practice. Moreover, there are theoretical results showing that for certain families of distributions such as discrete or Gaussian, violations will occur on a set of measure zero with respect to any continuous measures over parameters (Meek, 1995). However many argue that this should not be taken as a blanket assumption and that sometimes particular causal effects can occur precisely to balance out other ones, such as in for example biological systems or policy decisions (Hoover et al., 2001; Andersen, 2013).

Faithfulness is relevant in as if we consider the very general graphical model in Figure 2 to describe our scenario we can see by conditioning on $S$ we open up the paths $A \rightarrow X \leftarrow X^*(a)$ and $A \rightarrow S \leftarrow X \leftarrow X^*(a)$. Therefore if these paths are present and faithfulness holds we will have $X^*(a) \not\perp A \mid S = 1$ and so the counterfactuals cannot be captured by a model in $\mathbb{M}_{S=1}$. Therefore if it is assumed that the underlying data generating mechanism follows a model in $\mathbb{M}_{S=1}$, either for particular properties of causal DAGs or to approximate counterfactuals, justification should be given for one of the following, ordered by the strength of the assumption:

1. There is no selection from the general population; a census would satisfy this condition, for example.

2. There is selection from the general population, but the paths $A \to S \leftarrow X$ or $A \to X \to S$ are not present . This could be the case if we randomly sample individuals from the general population.

3. The selection depends on the covariates or sensitive attributes in such a way that the paths are present. However, it depends on them in such a way that violates faithfulness. In practice it is hard to see how to make a clear argument for this in fairness contexts as it would amount to saying that selection occurs in some suitable 'fair' way given by Proposition 3.

Now applying this to the running example of law school prediction we can see the law school applicants are a subset of the general population, so we violate 1. Further this selection is not random, and in general the likelihood of application would depend on GPA and LSAT so we also violate 2. Therefore in order to suppose a model in $\mathbb{M}_{S=1}$ that can fit the counterfactuals we have to make an argument for 3, but there is no obvious reason to suggest the data distribution would violate faithfulness. Therefore, there is no reason to believe structural counterfactuals from any model in $\mathbb{M}_{S=1}$ could capture the counterfactuals. Thus, by using these steps we have given a clear causal argument supporting the concerns we raised about the counterfactuals for the law school example in Section 3.

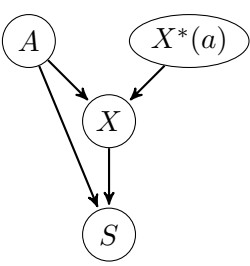

Figure 2: A causal DAG for selection

These steps can be applied to any dataset and if we plan on fitting a model in $\mathbb{M}_{S=1}$, justification for at least one of these points should be given. Unfortunately, in many fairness settings it is hard to imagine being able to argue for any of these, and so we run into the difficulties discussed at the end of the previous section. That is, we cannot generate both our dataset and the counterfactuals by models usually assumed in causal inference. Further, fitting any model to approximate the counterfactuals becomes significantly harder in practice.

### 4.3. Explicit Violation in certain cases

The scaled counterfactual fairness condition leads to a constraint that can be tested on a dataset to explicitly verify that no model in $\mathbb{M}_{S=1}$ can correctly capture the counterfactuals. It is worth noting that if this constraint is not clearly violated, that is *not* good evidence that a model in $\mathbb{M}_{S=1}$ does fit the counterfactuals, and instead the list of conditions for ignorability under selection in Section 4.2 should be referred to. The aim is instead to show beyond doubt that no such model fits. The constraint is as follows:

**Corollary 5** *If there exist $x, a, a'$ such that $P(X = x \mid A = a) > 0$ with*

$$P(S = 1 \mid X = x, A = a) > \frac{P(A = a \mid S = 1)P(A = a')}{P(A = a' \mid S = 1)P(A = a)}$$

*then there exists no model in $\mathbb{M}_{S=1}$ that has the correct counterfactuals.*

Note this places no restriction on the form of $\mathcal{M}^*$ apart from Assumption 1.

Table 1: Constraint for popular causal fairness datasets

| Does there exist an $x$ such that | Adult | Law school | German Credit |
|---|---|---|---|
| $P(S = 1 \mid X = x, A = \text{Female}) >$ | 0.475 | 0.753 | 0.421 |

We now apply this method to some datasets used in the causal fairness literature with sex as our sensitive attribute. The results are shown in Table 1. The bound is computed using census data to estimate the probability of individuals having a given sex in the general population.

We use the Adult dataset as an example to demonstrate the usefulness of this result; the observations in this dataset are taken from census data with certain deterministic constraints on the covariates. For example, the total number of hours worked has to be positive and the yearly earnings must be more than 100 dollars. Anyone satisfying these who is part of the census is guaranteed to get selected, and so for women with $x$ satisfying this $P(S = 1 \mid X = x, A = \text{Female}) = 1 > 0.477$. Therefore this violates the constraint and so there is no causal model in $\mathbb{M}_{S=1}$ that can accurately capture the true counterfactuals, regardless of the true causal model $\mathcal{M}^*$ that describes the world. A similar argument to the above can be applied whenever there is a deterministic rule for selection from the general population. Namely if $P(S = 1 \mid A = a) < P(S = 1 \mid A = a')$ and we can find a set of values for the covariates $X = x$ such that $P(S = 1 \mid X = x, A = a) = 1$, then the constraint is violated.

## 5. Challenges for Causal Fairness

In this section we discuss the challenges created when no model in $\mathbb{M}_{S=1}$ fits. We first look at the issues this causes to the general application of causal fairness methods and then to specific difficulties this creates for counterfactual fairness and path-specific counterfactual fairness.

### 5.1. Difficulties when no model in $\mathbb{M}_{S=1}$ fits

If no model in $M_{S=1}$ fits this does not mean that there exists no causal model which captures the true counterfactuals. However it does mean that in any correct causal model, the distribution of the noise variables will depend on $A$. This creates the following two challenges for the application of causal fairness methods.

Firstly the models that lie outside of $\mathbb{M}_{S=1}$ are not well behaved enough to guarantee many properties and identification results that are normally assumed in causal inference. The most obvious is that d-separation in the graph will not generally imply a conditional independence in the distribution. This is a problem as many key results in causality rely on d-separation, for example identification results, the do-calculus, and the adjustment criteria. Therefore if we find no model in $\mathbb{M}_{S=1}$ fits we should be cautious about applying results from the wider causal inference literature in fairness problems without clearly justifying that we still satisfy the required assumptions.

Secondly this makes the identification of counterfactuals strictly harder as we lose any way to connect them to real world observed variables. Under the assumption that some model in $\mathbb{M}_{S=1}$ fits we could identify the distribution of the group level counterfactuals without knowing the true model. That is, we know $P(X^*(a) \mid A = a', S = 1) = P(X \mid A = a, S = 1)$ using the assumption of ignorability under selection. Therefore we only need a way to find $P(X^*(a) \mid A = a, X = x', S = 1)$ in order to satisfy individual level counterfactual fairness. However if no model in $\mathbb{M}_{S=1}$ fits we are now in a strictly more challenging setting, where we cannot even identify the distribution of the group level counterfactuals. This is because the noise variables in our model are dependent on our sensitive attribute and the structure of the dependency is not clear without further assumptions or data. In the selection context this corresponds to the fact the distribution of those noise variable in the whole population is not identifiable from the data (Bareinboim et al., 2014).

This creates problems for fairness based on causal counterfactuals as we now need to introduce some dependencies between the sensitive attributes and any noise variables we include in our model. However this cannot be done arbitrarily as it is unclear how a particular dependency would affect the accuracy of your counterfactuals and therefore the fairness of the model. Thus, making any arbitrary changes to the model without any clear idea of its affect on the fairness would be high contentious. Furthermore making principled changes would require more assumptions, data or both.

## 5.2. Do Stuctural Counterfactuals from models in $\mathbb{M}_{S=1}$ have a causal interpretation?

We now ask, can we give the structural counterfactuals from a model in $\mathbb{M}_{S=1}$ a separate causal interpretation? Maybe as some other kind of counterfactual? We will argue not, and look at two possible interpretations. In the first the structural counterfactuals represent the counterfactual given that in the world where an individual is born with a different attribute they would have made it into the selected set. In the second it captures how an individual would appear at the time of selection if they counterfactually had a different sensitive attribute.

The first is in general difficult as we cannot identify who, if anyone, would have been selected in the real world and the counterfactual one in which they were born with a different sensitive attribute. This is related to work on causal inference in the presence of competing effects. Stensrud et al. (2020) explain competing events using the example of a 3 year medical trial and note that people in both the treatment and control group may die of other related effects before the trial is completed. As a result it is hard to come up with an appropriate counterfactual contrast for treatment effects as we cannot pinpoint who, if anyone, would have survived if they were in both the treatment and the control arm. Therefore the identification and definition of any counterfactual contrasts relies on strong untestable assumptions about a group of people who survive in both arms. In the same way it is hard to come up with the correct counterfactual contrast for fairness here as we cannot tell who, if anyone, would have made it to our selected set regardless of the value of their sensitive attribute at birth. This makes it challenging to take this interpretation and further to asses if we have achieved it.

In the second case, if we are trying to look at how an individual would appear if at the time of selection they had had a different sensitive attribute. The difficulty here is whether this is what we are aiming for: why should we propagate a causal effect through covariates that occur pre-selection? Again using the law school example, we could imagine saying we want our counterfactually fair predictor to align with the one in which an individual had a different sex in the moment of application. This seems to align with the intuition our predictor is fair if a female applicant will get in if the same applicant would be accepted if they were male. However if this is our aim, then it makes little sense to propagate causal effects through GPA and LSAT, as these occur before application. Instead the correct counterfactual would be obtained by simply flipping the value of our sensitive attribute, as Wachter et al. (2017) advocate for.

Therefore we argue that if ignorability is violated in our dataset violated then it is hard to believe that the structural counterfactuals from any model in $\mathbb{M}_{S=1}$ can be given a correct causal interpretation. This violation could either be due to a general rejection of ignorability (as in the case of race) or via the selection arguments in Section 4.

### 5.3. Counterfactual Fairness

In this section we consider what fairness guarantees are obtained when we aim to achieve counterfactual fairness with a model in $\mathbb{M}_{S=1}$ when none fits. In the previous section we argue that this will give no causal guarantees. Therefore the only fairness properties will be statistical. To this avail we provide the following:

**Proposition 6** *Let $\widehat{Y}$ be a predictor that is counterfactually fair according to some causal model $\mathcal{M} \in \mathbb{M}_{S=1}$. Then $\widehat{Y}$ satisfies demographic parity on the data it fits. That is $\widehat{Y} \perp A \mid S = 1$.*

This also leads to the following corollary which gives a more general relationship between counterfactual fairness and demographic parity when we make Assumption 1.

**Corollary 7** *Let $\mathcal{F}$ be the set of predictors, possibly with random noise, which are counterfactually fair according to the true model $\mathcal{M}^*$ under Assumption 1. Then we have that all $\widehat{Y} \in \mathcal{F}$ will satisfy demographic parity if and only if we have ignorability under selection.*

In other words, if we do not have the conditions for ignorability under selection then truly counterfactually fair predictors need not necessarily satisfy demographic parity.

Therefore we argue that in many fairness settings, applying a model from $\mathbb{M}_{S=1}$ with the hope of achieving counterfactual fairness will only give us demographic parity, with no extra causal interpretation.

### 5.4. Path Specific Fairness

Recently, new causal fairness variants have been proposed that rely on the use of path specific effects (Nabi and Shpitser, 2018; Chiappa, 2019). These involve labelling pathways from our sensitive attribute to outcome of interest as fair or unfair, and controlling for the effect along the unfair pathways in our predictor. We give a brief example of these methods using the classic UC Berkely Gender discrimination case. In this dataset we find

that there is a correlation between acceptance rate to Berkeley and Gender, which seems unfair. However after stratifying by department this correlation disappears. Therefore the argument is the original correlation is due to the fact that the women who apply, apply for more competitive departments.

In Chiappa and Isaac (2018) they represent this by the causal DAG in Figure 3 with covariates gender ($A$), department ($D$), qualifications ($Q$) and acceptance to Berkeley ($Y$). They argue that gender has two potential causal effects on outcome, a direct effect and an effect through department. They label the direct effect as unfair and the indirect effect as fair. Now according to path specific fairness the outcome is unfair if it takes in any causal effect in through unfair pathways, therefore by the same argument we have that a fair predictor will only take information about the sensitive attribute through fair pathways.

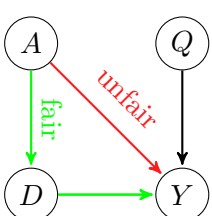

Figure 3: Causal DAG for the Berkeley Gender Discrimination

### 5.4.1. IDENTIFIABILITY OF PATH SPECIFIC EFFECTS

In order to be able to apply these path specific fairness methods we need to be able to identify the underlying path specific effects. In the Berkeley data, for example, we need to be able to identify the distribution of $Y(D(a), a')$, denoting the effect when $A$ is set to $a$ along the pathway leading to $D$ and $a'$ on the direct pathway to Y. In general results for the identification of path specific effects rely on the correct causal model having independent noise. Without this assumption we do not necessarily have identifiability of path specific effects (Shpitser, 2008). However if we assume an ancestrally closed sensative attribute set, this is just $\mathbb{M}_{S=1}$. Therefore, the conditions in Section 4.2 can also allow us to reason about when we can apply path specific fairness variants with such DAGs. Furthermore we can apply the results in Section 4.3 to the experiments on the Adult dataset in both Chiappa (2019) and Nabi and Shpitser (2018). As the constraint in Corollary 5 is violated, we do not have the necessary conditions to guarantee identifiability of the path specific effects these experiments rely on.

## 6. Conclusion

In this paper, we have argued that more care should be taken as to what type of SCM is assumed in order to achieve fairness through the use of structural counterfactuals. We show that often any SCM with independent noise cannot capture the correct counterfactuals and further that the structural counterfactuals from these models cannot be given a clear causal interpretation. Finally we have show this creates issues for a variety of causal fairness methods due to model fitting and an inability to apply results from the wider causal inference literature.

## Acknowledgments

We would like to Siu Lun Chau, Jean-Francois Ton and the reviewers for their helpful comments that have greatly improved this paper. Jake Fawkes also gratefully acknowledges funding from the EPSRC.

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
