# OpenReview forum: "Selection, Ignorability and Challenges With Causal Fairness "
_cclear.cc/CLeaR/2022/Conference — CLeaR 2022 Oral_

### Official Review · Reviewer_V4FC · 2021-11-11

**Confidence:** 3
**Overall Score:** 8

**Main Review:**

I think that this paper is a valuable contribution, because it has a critical look at methods used in practice. Moreover, the authors provide concrete theoretical results which could help identify suitable scenarios of application in practice. The proofs in the Appendix are nicely written and rather easy to read.
To aid the practitioner, I would suggest to include the numerical details (probably in the appendix) for one running example. E.g. how would Prop 3 be used in such an example and how are the details in corollary 4 computed in this example (e.g. otbtaining 0.477).
Apart from this I think this is a valuable paper and I only have minor comments:
-	p. 1: “only be understanding”, be => by
-	p. 3: “the fact the model”=> “the fact THAT the model”
-	p. 3: Define “SCM” when you mention structural causal models
-	p. 3: “it valid” => “it is valid”
-	p. 6: “we have access too” => “we have access to”
-	Def. 1: “to be the” => “be the”
-	P. 10: “define define”
-	p. 12: “same applicant would be accepted were accepted” => use only one of those expressions


**Summary:**

In this paper, the authors have a critical look at popular fairness methods based on causal counterfactuals. They reframe the problem as one of selection from a wider population and come up with conditions and a testable constraint for the existence of a causal model with independent noise that has the correct counterfactuals. The approach is illustrated using three examples. Finally, the authors apply their findings to path specific fairness and discuss the interpretation of fitting results if ignorability under selection is not given.

---

> ### Author Response · Authors · 2021-12-03
> **Reply to Reviewer V4FC**
>
> We thank the reviewer for their review, comments and proposed
> changes. Following these comments we will restructure the paper by moving Section 4 into
> Section 3 as a new subsection, after Proposition 3. We do this
> because Section 4.1 provides details of how to reason about
> potential violations of Propositions 2 and 3 in practice, as well as
> discussing these steps in relation to the running example of law
> school admissions. Further, Corollary 4 builds upon Proposition 3
> and so should be included as an extension of how a practitioner
> may think about Proposition 3. We hope that by moving these
> Sections immediately after Proposition 3 we will make clear their
> relationship to how a practitioner should think about these results.
> Finally following similar comments also made by Reviewer L3a9
> about the lack of detail in in Section 4.2 about computations as well
> as the datasets used we intend to add a new appendix on this topic including detailed calculations of this bound as well as descriptions of the datasets.

---

### Official Review · Reviewer_iM9N · 2021-11-19

**Confidence:** 2
**Overall Score:** 7

**Main Review:**

I would first like to note that my expertise is in selection and ignorability, not causal fairness, so it is very likely that many of my thoughts will be off-the-mark due to my blind spots.

I generally liked the paper and the point it is trying to make. The idea that the data we have access to is selected, and pretending otherwise can lead us to fundamentally incorrect results, is important. This also has come up in economics, where some of Roland Fryer’s work on police discrimination has been criticized for ignoring selection into police interaction (https://statmodeling.stat.columbia.edu/2016/07/14/about-that-claim-that-police-are-less-likely-to-shoot-blacks-than-whites/). This is an important point which the authors do well to formalize.

I expect this paper is significant. Based on the paper's existence and its references, I assume that selected data is an under-addressed worry in the fairness literature. It is possible that topic experts would disagree about the claim's novelty, but if this insight is relatively novel, then it is important.

My main critique is around a lack of formalization where it would be useful in the setup. The authors seem to have in mind a scenario where a “prediction” (“which our predictor should not be discriminatory on the basis of”) is equivalent to a “decision” (“a decision to grant”), and both of these are related to an underlying causal model. I could imagine a decision made based on a prediction formed from a causal model. But I could also imagine a prediction meant to predict new data from the observed model. Or a decision rule existing independently of the model. All these terms were floating. I would have appreciated a formal definition of the setup, i.e. what we receive as data, what objects are given independently of the data, what objects are given as a fixed function of the data, what objects we can control in response to that data, what the domains of these objects are, and what null we wish to test. In this absence, I had to infer what these terms are meant to represent from the discussion, and perhaps led to needless muddy thinking on my part.

My next substantial critique is that the authors under-sell how we should think about ignorability in their setting. The authors seem to pitch ignorability as a restriction on what they can cover that is satisfied in cases like sex and certain disabilities. I think this is too pessimistic and wrong on both counts. First, I think it would be better to think of global population ignorability as a best-case. As the authors acknowledge, there are many fairness attributes, like race, that are clearly not going to be ignorable in any plausible dataset. Assuming ignorability makes our task easier. Thus I read this paper as proceeding in the most favorable circumstance for the analyst. Second, the authors’ examples of ignorable attributes are at best likely to be nearly ignorable. For example, even if we think of fetal sex as randomized with constant probabilities, later abortion, adoption, and the number of subsequent sibling births can be a function of that realized sex. It is therefore unlikely to be ignorable in practice. Thus, once again, I read this paper as proceeding after conceding an unreasonably favorable assumption to the practitioner. This paper ought to take more credit.

I have a suggestion for follow-up work that thinks about how to proceed in cases where ignorability fails (either at the population level or through selection) but perhaps nearly holds. There is a large literature on treatment effect estimation with non-ignorable selection (e.g https://arxiv.org/abs/1808.09521 and https://rss.onlinelibrary.wiley.com/doi/pdf/10.1111/rssb.12327). There is also work on generalizing RCTs after selection on observables (https://www.ncbi.nlm.nih.gov/pmc/articles/PMC4359056/ and https://arxiv.org/abs/2111.01357). I understand Melody Huang is extending that work on RCTs on selected data to selection on unobservables. This is a setting where the authors imagine their task as selected data after an RCT. Perhaps similar ideas could be extended.

I then have smaller critiques:

(1) I am not convinced that “dependent noise” is a useful framing in this setting. The distribution of unobservables seems like at best a useful device for learning something tangible, but I personally have never found fictitious unobservable variables themselves to be a good way to think about confounding. I find potential outcomes to be the most tangible, so that statements about independence are almost coherent to think about, and selection probabilities nearly as tangible. I don’t see what the unobservable and intangible noise variables themselves add to our understanding beyond those more tangible objects, but perhaps other readers feel otherwise and my view in this respect ought to be discarded

(2) Relatedly, I am not sure why dependent noise variables are more troubling than the idea that we’d need to specify an exact selection mechanism to recover point-identified counterfactuals

(3) It likely reflects my lack of knowledge, but I would have appreciated formal definitions of the “faithfulness assumption” and “demographic parity"

(4) A formal definition is especially important for faithfulness, which Section 4.1 offers as the only out for meaningfully selected data. My best guess was that it meant the population DAG does not generate the data so that we need to specify a selection mechanism to recover counterfactuals. Perhaps it is something else

(5) I did not see how the abstract’s discussion of demographic parity (“it aims to replace”) was supported by the rest of the text, but it may just be common knowledge to practitioners

(6) I did not find the “only if” proof for Proposition 3, though I anticipate it to be a quick Rosenbaum and Rubin proof and I may have simply misread

(7) I think the right-hand side of Corollary 4 would be clearer as P(S = 1 | A = a) / P(S = 1 | A = a’)

(8) “Structural” has a meaning in economics (estimating parameters of a true DGP that we cannot imagine changing) that is in the vicinity of an antonym to the meaning here (estimating parameters of a structural causal model that may be incorrect). It has also created a hellscape of different understandings of the term. I suggest the authors learn from our errors and use a less ambiguous term like “SCM-based”

I did not feel like I had the expertise to evaluate Section 5 (path specific fairness). I leave it to others to judge whether that section is useful.

**Summary:**

This paper considers studying decision fairness. As I understand it, the paper considers a setting where we believe a protected attribute is ignorable and we are interested in whether observed decisions would have the same distribution regardless of protected attribute. The paper points out that when we only have access to decisions on a subset of the population, e.g. law school decisions for applicants, then this notion of fairness can (outside of a special case) only be true in both the global population and the subset with data if selection into the subset is independent of the sensitive attributes and the included covariates.

---

> ### Author Response · Authors · 2021-12-03
> **Reply to Reviewer iM9N**
>
> We thank the reviewer for their insightful and detailed review. The points made and further resources suggested are both interesting and helpful. In particular we will add Steven N. Durlauf and James J. Heckman's critique of Roland Fryer's work (https://www.journals.uchicago.edu/doi/full/10.1086/710976?casa_token=gjhjHY2SMYEAAAAA:hsFRha3YO1UcBhizCjz6mjr2zuIkzcJhRcan0wk0e5pm_yhRhsa7LGu0H0Y2-CsuafivIKkSlQ) as a reference for the selection portion of the paper as we find this to be an interesting comparison that strengthens the point. Our response to the critiques is as follows:
> The point on the lack of formalism for definitions and the setup is a helpful one. It is similar to some points raised by Reviewer 3Eg1 asking for more clarification on notation in Section 2.1. In response to both of these comments we will be improving the clarity of our notation and giving clearer definitions of the set up with the hope that it will make the paper easier to read and understand. With regards to distinguishing between predictions and decisions, we intend only to consider fairness of a predictor.  We appreciate that by using the word `decision' in the abstract as well as in the line referenced by the reviewer we have created confusion.  We will remove all references to decisions. We will further add-in to the set up the point about trying to predict some outcome of interest, $Y$, as this is what the causal fairness literature mostly has in mind (not prediction of new data).
>
> We appreciate the point the reviewer makes with regards to the strength of ignorability. Following these points we will place more emphasis on ignorability by giving it its own Section in which we detail it more. Further we had not made it clear enough that the main thing this paper is taking issue with the assumption of ignorability in fairness contexts. We will emphasise that for many sensitive attributes such as race it is hard to imagine any dataset in which these traits are even approximately ignorable. The selection argument that then follows is aimed as a rejection of ignorability in fairness settings only for sensitive attributes like birth sex, which we can imagine as approximately ignorable in some populations. We thank the reviewer as we feel these changes will make the argument clearer and stronger.
>
> Other Comments:
> 	1)	We appreciate the reviewer's thoughts and opinions on this, however it feels more like a challenge to Pearl's formalism of causal and counterfactual inference. We wrote everything in this manner because it is used in the original counterfactual fairness paper, and (we think) makes it easier for readers to follow our arguments. Further, as stated the original paper requires taking some of these unobserved variables as inputs, so the variables themselves and a distribution over them are directly relevant.
> 	2)	For the case of selection, an exact selection method would do the same thing. However, the point is that dependent noise is required due to a rejection of ignorability when it comes to, for example, race. There is no selection mechanism here and so dependent noise captures both. We will move Section 3.1 out of Section 3 to make it clear it relates to more than selection.
> 3/4) Similar point to reviewer Reviewer L3a9. We will define both clearly.
> 5) This is more supported by the original paper. It seems this Section is not clear/detailed enough from this comment and that of Reviewer L3a9. We will alter this.
> 6) Apologies, in fact the proof of equivalence is just a trivial additional line as we already have that the conditional independence is equivalent to the first equation, which is in turn equivalent to counterfactual fairness (all under assumption 1).
> 7) We prefer to leave it as it is as there is a non-trivial (although common) cancellation when $P( S=1 | A=a)/P(S = 1 | A = a')$ is expanded by Bayes' rule. This probability is much easier to evaluate than something like $P( S=1 | A=a)-P(S = 1 | A = a')$ as a result, especially when the probability of selection is very small. We feel it is clearer to not require the reader to do this evaluation in their head, or even worse to look at the bound and decide it is too hard to practically evaluate.
> 8) Thank you for this. It is worth noting that we did not introduce this term and it has also been used in: https://arxiv.org/pdf/2002.06278.pdf. SCM-based does seem to create less confusion however, so we will change it.

---

> > ### Comment · Reviewer_iM9N · 2021-12-08
> > **Three Clarifying Questions Re: Comment**
> >
> > Three very small questions to make sure I am understanding your comment:
> >
> > (1) I’m not sure I’d agree that your paper is “taking issue with the assumption of ignorability in fairness contexts.” I read it as taking issue with assuming ignorability under selection (or at least showing how strong of an assumption that is). Was that a misreading on my part?
> >
> > (2) What do you mean by “There is no selection mechanism here?”
> >
> > (3) Is "structural" the standard notation in the fairness literature? If so, you should probably align with the literature rather than this isolated reviewer
> >
> > Thanks! And thank you for the thorough reply!

---

> > > ### Author Response · Authors · 2021-12-15
> > > **Re: Three Clarifying Questions**
> > >
> > >
> > > We thank the reviewer for further clarifications and continued feedback.
> > > 1) The selection argument we presented only follows for sensitive attributes for which we are happy with assumption 1. So traits that are (approximately) ignorable in the general population. We did try to make the point that for some attributes such as race you wouldn’t feel happy making the assumption of ignorability in any population and so assumption 1 is unrealistic. However this does not mean for something like race this paper doesn’t apply and that the practitioner should be comfortable with assuming independent noise and ancestral closure, as many of the references we cite do. In fact it means the practitioner should not be comfortable with this in any dataset, even if it was for the whole population. This is why we feel it is not just ignorability  under selection (even though this is obviously a large part of the paper). If any of this was unclear or remains unclear we would appreciate the reviewers feedback.
> > > 2) This relates to the previous point as because we do not have ignorability  in the whole population we would still be unsatisfied with the independent noise assumption (with ancestral closure). This is despite the fact that in the whole population there is no selection. The lack of selection here is what we mean by no selection mechanism but still rejecting independent noise.
> > > 3)The paper we references is the only mention we can find of this term in the fairness literature however there are other mentions of it by for example Pearl in: http://ftp.cs.ucla.edu/pub/stat_ser/r413-reprint.pdf. We will consider further about which terminology to use as whilst the previous term has been used a few times, SCM based does make clear that these depend on a specific SCM.
> > >
> > > Edit: Fixed typos caused by autocorrect on ignorability

---

### Official Review · Reviewer_L3a9 · 2021-12-01

**Confidence:** 3
**Overall Score:** 8

**Main Review:**

I found this paper to be an interesting & high-quality read, and found the references to be helpful (and indicative of the significance of this work). Some minor commentary below:

Broader points:
- It would be helpful to rigorously define the faithfulness assumption when it is first introduced, to provide more context here:  “if the general population satisfies a particular Markov structure, the faithfulness assumption inherent to directed graphical models suggests that the selected population will not. Hence, the noise variables effectively must be dependent.”  It would also be helpful to include a law school example of potential unfaithfulness wherein two effects could balance and cancel.  Similarly would be helpful to explain in words why “If we condition on S=1 then we open up a path between A and X*(A)”.
- It could be nice to include more discussion on unobserved covariates and how that affects this framework: e.g., what about the latent variable for “Knowledge” in Kusner paper?
- It may be helpful to bring up the data censoring issue (described nicely in the context of competing effects) earlier on as well -- in section 2 -- when describing the ways in which the law school counterfactuals might not hold.
- I found the final paragraph of section 4.2 to be a bit lacking (e.g., ‘Adult / German Credit’ are not described, the 0.477 is not contextualized, etc.) -- more details here could help.
- I assumed the demographic parity connection would play a larger role in the paper given its mention in the abstract, but the content in section 6.2 was quite short and I’m not sure I gathered intuition for why this is the case.  Perhaps consider removing from the abstract, or expanding in 6.2.

Smaller nits:
- When first referencing SCM, it may be helpful to refer to the full “structural causal model” name
- Worth including a footnote with further description of what the Adult dataset entails. (Likewise for German Credit, which doesn’t appear anywhere except Table 1.)
- Define evidence E before its first appearance, not after
- Would be helpful to clean up terms like “evidence”, “contexts”, and “measured covariates” and simply refer to them all consistently as “observed covariates”
- Applying the Bender Rule here: Worth mentioning where the law school example is specific to (e.g., the US, Canada) to contextualize the social concerns brought up in this paper
- P. 5 “Considering this in the context…” -> “Consider”
- Footnote 4, grammar needs fixing
- Section 6.2 “demographic parity” is missing a period
- Appendix B first sentence, grammar needs fixing
- Appendix G: “Lemma ??” LaTeX error


**Summary:**

This paper observes that the ignorability assumption is poorly applied to causal fairness applications, resulting in unrealistic counterfactuals (e.g., recognizing that counterfactuals to men applying to law school are not comparable to “true” women applying to law school).  If we have ignorability under selection as defined, we can achieve counterfactual fairness when fitting a model using samples selected from a larger population; otherwise, the noise distribution will be dependent on the sensitive attribute.

---

> ### Author Response · Authors · 2021-12-03
> **Reply to Reviewer L3a9**
>
> We thank the reviewer for their thoughtful review and comments about the paper. Our response to the suggestions is as follows:
> 	▪	Following comments given by this reviewer as well as Reviewer iM9N we will include the definition of faithfulness. We can explain non-mathematically what faithfulness means, give examples of violations in other contexts; however, stating it in the law school example would be challenging because it seems very implausible that such a condition would arise--- this is a strength of our argument. In general we feel that the onus should be on those supporting this framework of counterfactual fairness to argue such violations could even exist in practice, and perhaps to give examples.
> 	▪	 We note that the definition we gave for SCMs was broad enough to include this latent variable model of Kusner, as we allowed the same ‘noise' variable to be a parent of multiple observed variables. We will make this more clear.
> 	▪	 This data censoring issue was spoken about less technically in Section 2.3, where we discussed the fact that an individual may not have attended college to get a GPA if they had counterfactually had a different sensitive attribute, and so applied to law school. We will make this connection more explicit though.
> 	▪	 This relates to comments made by Reviewer V4FC and, as we have mentioned previously, we will add an appendix with further details about these datasets, the calculation of these values; we will also include a brief description of the Adult dataset.
> 	▪	 From our perspective the whole of Section 6 relates to demographic parity. This is a two stage argument in which we first discuss causal interoperation, for which we argue there isn't any, therefore the only guarantees are statistical and are just equivalent to demographic parity.  However, following this discussion as well as Reviewer 3Eg1's comments on Section 5 we intend to merge Sections 5 and 6 (with possible reorderings) to give a whole Section discussing the challenges we raise with these causal fairness methods. This Section will then be referenced in the abstract to reduce emphasis on these results, which make up only a small section of the paper.
>
> Finally we thank you for all your additional comments, specifically the Bender Rule. We will implement these changes.

---

### Official Review · Reviewer_3Eg1 · 2021-12-01

**Confidence:** 3
**Overall Score:** 7

**Main Review:**

I found the draft really interesting. It is quite profound in challenging commonly assumed conditions in counterfactual fairness literature. I am super familiar with causal inference but not with fairness. Personally, I found the paper very useful to gain intuition and to learn counterfactual fairness.

I have some minor comments:
1. I found section 5 a deviation from the topic. I am not sure if section 5 is that essential in this draft, I felt it is an example, which really exhausted a lay person like me to read because I am not familiar with it. Can you de-emphasize it, maybe move to appendix (if you can), or move it after section 6 which I found more interesting.

2. I think sections 1 & 2 are well written and the authors are clear with their goals. But sections 3 and 4 start to become vague and dry. The notation becomes heavier and too much English took me long time to digest. Can you put more notation back in section 2.1 when you introduce notation so readers can easily find a ''dictionary'' to know the notation and do not need to skim through the paper to find notation?

**Summary:**

The paper challenged commonly adopted assumptions in counterfactually fairness papers. The authors explained when the assumptions can be questionable and gave sufficient conditions to reject the assumptions.

---

> ### Author Response · Authors · 2021-12-03
> **Reply to Reviewer 3Eg1**
>
> We thank the reviewer for their suggestions and comments.
>
> 1. Whilst we think this Section is important, as it provides a novel critique of some of the more recently popular causal fairness definitions, we do agree that in its current position it disrupts the flow of the paper and may be overly technical. As such we will merge Sections 5 and 6 into one larger Section on challenges with causal fairness with the main emphasis being on Section 6. Finally we will move some of the details/definitions to the appendix and describe it in a less technical way that is more clearly understandable to someone unfamiliar with path specific effects.
> 2. This point feels like it relates to that raised by reviewer iM9N about the lack of formalisation and definitions of certain things. As such we will be making various changes to include the clarity of definitions in Section 2.1 and giving explicit definitions of anything used in Sections 3 & 4 so that this may be used as a dictionary.

---

> > ### Comment · Reviewer_3Eg1 · 2021-12-06
> > **Follow up comments**
> >
> > One more thing to add, the main motivation is that the existing literature assumed that the sensitive attributes are ancestrally closed. Also, the authors have given a brief summary of the literature assuming it on page 6. But after I looked into those literature, I found it hard to find where they assumed it except Kusner et al (2017). Can you elaborate on this part since this motivates the study? Also, in Kusner et al (2017), it seems like ancestral closure is not the main assumption, is it?

---

> > > ### Author Response · Authors · 2021-12-09
> > > **Response to the follow up comments**
> > >
> > > We agree that this assumption is not explicitly stated in any of the literature apart from Kusner et al however all the papers we cite on page 6 make this assumption for all examples they give. We note that the reason ancestral closure is not discussed in the other papers on page 6 is not because this assumption isn’t made, but because it relates more to the construction of DAGs than to any specific causal fairness method. As such it is not specific to causal fairness. Following this any examples we were able to find in the wider literature of a DAG containing nodes like race and sex we found them to be without parents and so the set of sensitive attributes to be ancestrally closed. So it is not the case that any of these models explicitly have this as a required assumption, but all examples we have been able to find assume this in their DAG. Therefore we feel it is not a great restriction at all to only look at models with ancestrally closed sensitive attribute set.
> > >
> > > Kusner et al (2017) is a bit different as they point out if you don’t have an ancestrally closed sensitive attribute set then counterfactual fairness makes less sense. They give the example of mother’s race and race where if mothers race is not also sensitive we can make a predictor that discriminates on the basis of mothers race and it would still be counterfactually fair with respect to race. Again this is not the main assumption of the paper but it does motivate ancestral closure in a fairness context as otherwise we can discriminate on the basis of something that causes our sensitive attributes and still be counterfactually fair.
> > >
> > > We hope this answers any questions on this. If the reviewer feels any of this is unclear in the original paper we are happy to add more clarification to this section.

---

### Author Response · Authors · 2021-12-03
**Response to all Reviewers**

We are very grateful to all the reviewers for their detailed, insightful reviews and very helpful comments.

---

### Decision · Program_Chairs · 2022-01-12

**Decision:**

Accept (Oral)

**Comment:**

Thank you for your submission to CLeaR and engagement with reviewers during the discussion period.

This paper analyzes the use of causal counterfactuals in fair decision making, finding that achieving fairness requires correctly taking into account ignorability under selection. Reviewers appreciated the insight and clarity provided by this paper, its concrete theoretical results, and in particular the importance and relevance of its critical analysis of popular fairness methods, illustrating their shortcomings when specified conditions are not met.

We hope the detailed reviews are useful and encourage authors to incorporate the clarifications provided in their responses to reviews within a camera-ready.